# ZFR coordinates crosstalk between RNA decay and transcription in innate immunity

Nazmul Haque[1], Ryota Ouda[2], Chao Chen[2], Keiko Ozato[2] & J. Robert Hogg [1]

Control of type I interferon production is crucial to combat infection while preventing deleterious inflammatory responses, but the extent of the contribution of post-transcriptional mechanisms to innate immune regulation is unclear. Here, we show that human zinc finger RNA-binding protein (ZFR) represses the interferon response by regulating alternative pre-mRNA splicing. ZFR expression is tightly controlled during macrophage development; monocytes express truncated ZFR isoforms, while macrophages induce full-length ZFR to modulate macrophage-specific alternative splicing. Interferon-stimulated genes are constitutively activated by ZFR depletion, and immunostimulation results in hyper-induction of interferon β (IFNβ/IFNB1). Through whole-genome analyses, we show that ZFR controls interferon signaling by preventing aberrant splicing and nonsense-mediated decay of histone variant *macroH2A1/H2AFY* mRNAs. Together, our data suggest that regulation of ZFR in macrophage differentiation guards against aberrant interferon responses and reveal a network of mRNA processing and decay that shapes the transcriptional response to infection.

[1] Biochemistry and Biophysics Center, National Heart, Lung, and Blood Institute, National Institutes of Health, 50 South Drive, Room 2341, Bethesda, MD 20892, USA. [2] Division of Developmental Biology, National Institute of Child Health and Human Development, National Institutes of Health, 6 Center Drive, Room 2A01, Bethesda, MD 20892, USA. Correspondence and requests for materials should be addressed to N.H. (email: nazmul.haque@nih.gov) or to J.R.H. (email: j.hogg@nih.gov)

The innate immune system combats infection by rapidly reprogramming cellular gene expression. The response to infection is initiated by recognition of unique features of molecules produced by pathogens (known as pathogen-associated molecular patterns (PAMP) by specialized cellular pattern recognition receptors (PRR)[1]. PRRs such as RIG-I/MDA5-type RNA helicases detect common features of viral mRNAs, 5′ triphosphates or extended double-stranded RNA structures, whereas Toll-like receptors (TLR) recognize a range of macromolecules commonly produced by pathogens[2–4]. Activation of PRRs results in transcriptional induction of cytokines including interferon-α (IFNα) and IFNβ, driving expression of a large network of interferon-stimulated genes (ISG) with diverse functions in inhibiting the spread of pathogens[5]. Innate immune responses can arise from many cell types throughout the body, including cells not typically thought of as immune cells, such as fibroblasts and endothelial cells. In addition, specialized cells including macrophages and dendritic cells function as professional sentinels primed for cytokine production upon pathogen detection.

Studies of gene expression in the innate immune system have focused primarily on transcriptional regulatory events. However, post-transcriptional regulation, including alternative pre-mRNA splicing, is now thought to be essential for controlling tissue-specific and condition-specific gene expression programs[6–8]. Alternative splicing can alter protein functions by changing mRNA coding sequences or can be coupled to additional downstream post-transcriptional regulatory mechanisms to provide additional layers of control over protein expression and localization. For example, alternative splicing is frequently coupled to nonsense-mediated mRNA decay (NMD) through regulated inclusion or skipping of exons encoding premature stop

codons. This mechanism allows independent regulation of transcriptional rates and protein production, and is active in many important cellular processes.[9]

The highly conserved zinc finger RNA-binding protein (ZFR) is essential for mouse early embryonic development and mutated in the human neurological disorder hereditary spastic paraplegia[10–12], but its physiological and biochemical functions are mostly unknown. ZFR comprises three N-terminal C2H2 zinc finger motifs and a C-terminal DZF domain, a conserved nucleotidyltransferase fold lacking essential catalytic residues (Fig. 1a)[12]. Structural studies suggest the ZFR DZF domain could be used to heterodimerize with the DZF domains of multifunctional RNA-binding proteins ILF2/NF45 and ILF3/NF90, while the zinc fingers probably engage RNA targets[13]. Notably, all three zinc fingers are conserved from *Caenorhabditis elegans* to humans, indicating strong selective pressure to maintain specific interactions.

Here we show that ZFR is a potent regulator of alternative splicing, with an important role in preventing excessive type I interferon activation in multiple cell types including human monocytic THP-1 cells and primary bone marrow-derived macrophages (BMDM). Consistent with an important role for ZFR in macrophages, distinct ZFR isoforms are expressed in monocytes and macrophages; monocytes express truncated ZFR, whereas macrophage differentiation causes a switch to full-length ZFR expression. In the absence of ZFR, many differentiation-induced splicing changes are attenuated, implicating ZFR as a major post-transcriptional regulator of macrophage-specific gene expression. Furthermore, we find that ZFR promotes expression of the histone variant macroH2A1 (mH2A1) to suppress aberrant type I interferon production. THP-1 cells depleted of ZFR exhibit constitutive elevation of ISG expression and hyperactivation of

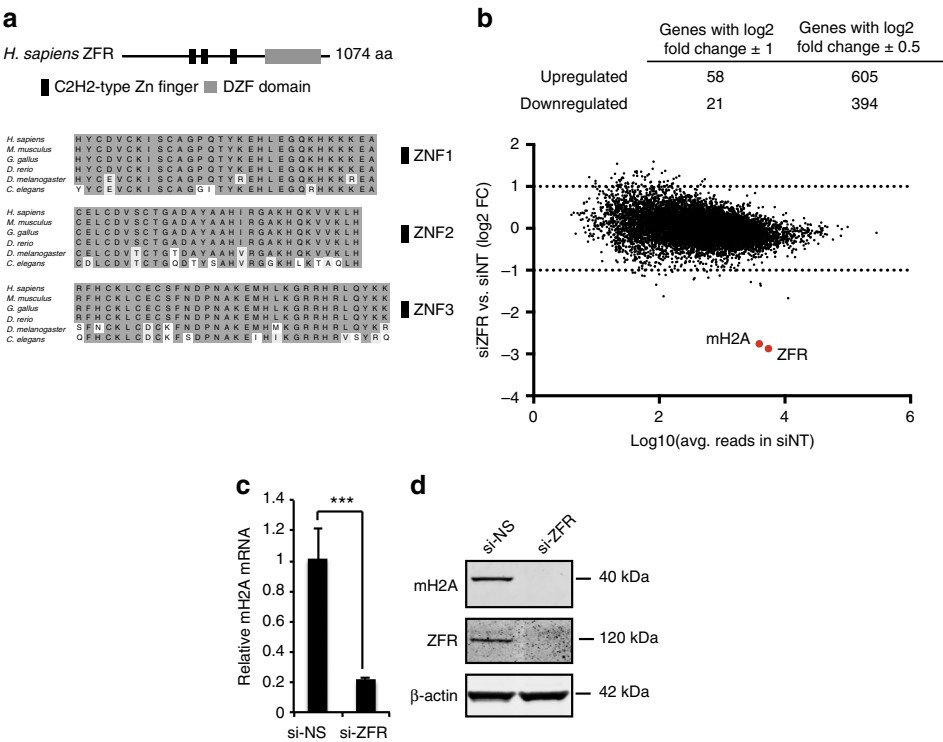

**Fig. 1** Evolutionarily conserved ZFR protein is required for macroH2A1 expression. **a** Schematic of the ZFR protein depicting N-terminal Zn-fingers and C-terminal DZF domain (top), and sequence conservation of ZFR Zn-fingers in the indicated species (bottom). **b** RNA-seq analysis of HEK-293TO cells transfected with si-ZFR or si-NS (n = 3). Top, numbers of genes upregulated or downregulated by ZFR depletion at the indicated thresholds, with adjusted P < 0.05. Bottom, scatter plot of fold change in mRNA abundance vs. number of reads derived from each gene, with *mH2A1* and *ZFR* highlighted in red. **c** qRT-PCR analysis of mH2A1 in cells treated as in **b**. Graphs indicate mean ± SD; ***P < 0.001 (two-tailed Student's t-test). **d** Immunoblot analysis of ZFR and mH2A1 in cells transfected with si-ZFR or si-NS as in **b**. β-actin was used as a loading control. Approximate MW based on size markers is shown

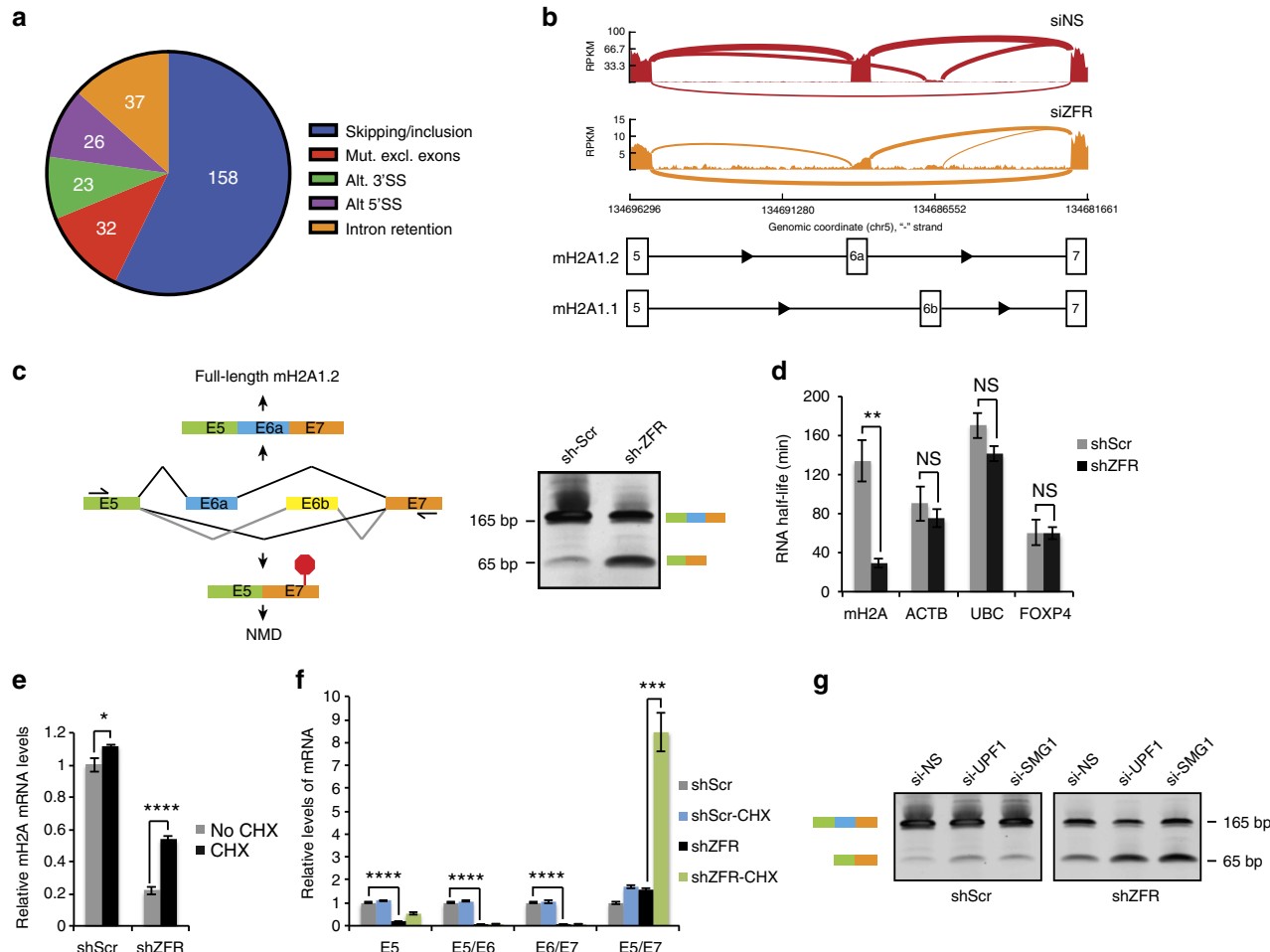

**Fig. 2** ZFR regulates alternative splicing and decay of mH2A1 mRNAs. **a** MISO analysis of alternative splicing in ZFR knockdown cells. Numbers of alternative splicing events of the indicated types identified with bayes factor > 20 and change in isoform usage of $|\Delta\Psi| > 0.12$ are shown. **b** Sashimi plot illustrating skipping of mH2A1 exons 6a (mH2A1.2 isoform) and 6b (mH2A1.1 isoform) in ZFR knockdown cells. RNA-seq read densities from representative replicates are shown; note that ZFR RNAi plot is re-scaled to allow visualization of the low levels of *mH2A1* mRNA. **c** Left: schematic depicting incorporation of a PTC upon skipping of both E6a and E6b of *mH2A1* pre-mRNA, generating a potential NMD substrate. Right: RT-PCR with primers spanning differentially included E6a of *mH2A1* RNA. HEK-293TO cells were stably transduced with the indicated shRNAs; schematic depicts long (top band) and short (bottom band) isoforms. **d** Half-lives of indicated mRNAs were determined by metabolic labeling of newly synthesized RNA ($n = 3$) in cells stably transduced with shZFR or shScr. **e** qRT-PCR analysis of *mH2A1* with and without treatment with cycloheximide (CHX; $n = 3$). **f** qRT-PCR as in **e**, performed with primers specific to the indicated *mH2A1* exon–exon junctions. **g** RT-PCR with primers spanning E5–E7 of *mH2A1* RNA from HEK-293TO cells transfected with siRNAs specific to UPF1 (si-UPF1), SMG1 (si-SMG1), or non-targeting control siRNA (si-NS); schematic on left depicts long and short isoforms of mH2A. Approximate MW based on size markers is shown. Graphs indicate mean ± SD; *$P < 0.05$, **$P < 0.01$, ***$P < 0.001$, ****$P < 0.0001$ (two-tailed Student's *t*-test)

*IFNB1* transcription in response to immunostimulation. Identifying the mechanism by which ZFR suppresses innate immune signaling, we show that aberrant interferon activation in ZFR-depleted cells results from loss of mH2A1 expression. In the absence of ZFR, *mH2A1* pre-mRNA is aberrantly spliced and degraded by NMD, reducing mH2A1 protein levels. Moreover, we demonstrate that mH2A1 binds and represses the *IFNB1* promoter, accounting for enhanced *IFNB1* transcription in the absence of ZFR. Thus, ZFR is a link between transcription and post-transcriptional regulation of the type I interferon response by controlling the splicing and decay of *mH2A1* mRNA.

## Results

**ZFR is required for expression of histone variant macroH2A1.** We first investigated the function of ZFR in human gene expression by performing RNA-seq studies on HEK-293 cells after ZFR depletion (Fig. 1b; Supplementary Fig. 1A,

Supplementary Data 1). Comparison of transcript levels in cells treated with ZFR-specific siRNAs and non-silencing control siRNAs revealed modest changes in gene expression, with mRNAs from 118 and 43 genes upregulated or downregulated more than twofold, respectively.

Transcripts encoding the histone variant mH2A1 were striking exceptions, as they decreased by more than fivefold in response to ZFR knockdown (Fig. 1b). Corroborating the RNA-seq results, *mH2A1* mRNA expression decreased by ~80% in qRT-PCR experiments (Fig.1c), with a similar loss of mH2A1 protein (Fig. 1d). Three different shRNAs against ZFR had comparable effects on *mH2A1* mRNA (Supplementary Fig. 1B), and expression of a siRNA-resistant ZFR cDNA-rescued *mH2A1* mRNA in cells treated with ZFR siRNAs (Supplementary Fig. 1C-E), confirming that the disruption of mH2A1 expression was a consequence of ZFR depletion rather than siRNA off-target effects.

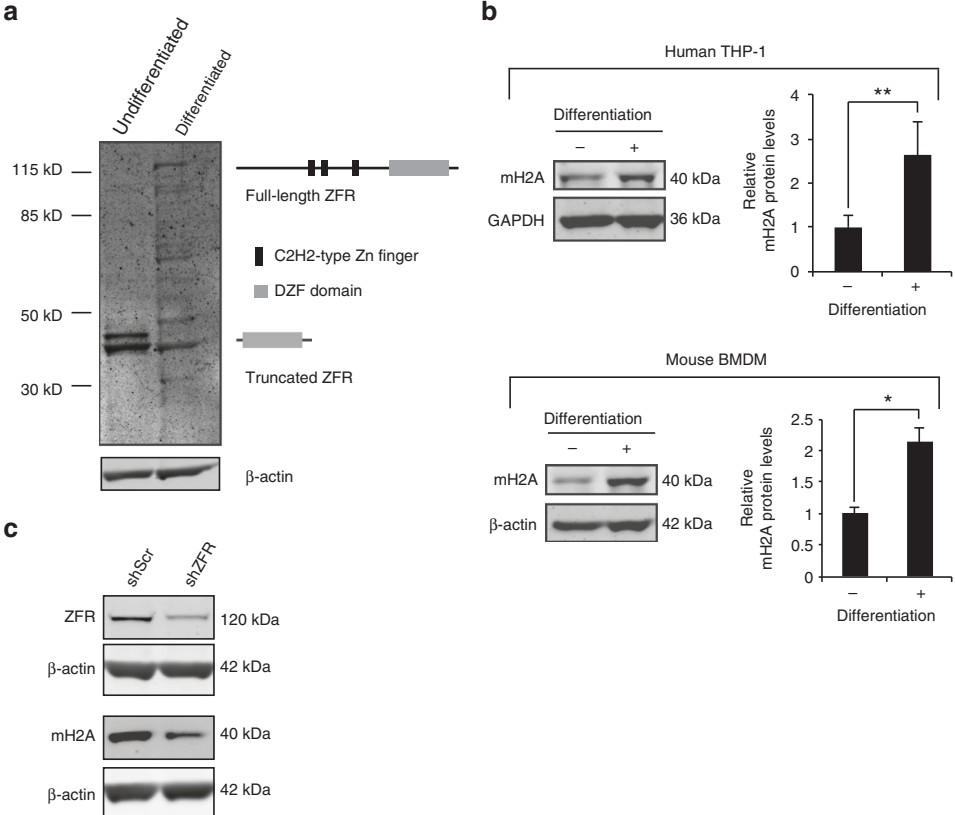

**Fig. 3** Regulation of ZFR and mH2A1 in macrophage differentiation. **a** Immunoblot analysis of alternative ZFR isoforms expression in monocytic and macrophage-like THP-1 cells, using an antibody against the C-terminal DZF domain. Schematics illustrate putative proteins generated by full-length and truncated ZFR isoforms arising from alternative promoters. **b** Immunoblot analysis of mH2A1 in human THP-1 cells (top) and murine BMDMs (bottom) before and after differentiation. Graphs represent quantification of mH2A1 protein levels in human THP-1 cells ($n = 3$) and mouse BMDMs ($n = 2$), with GAPDH as a control. **c** Immunoblot analysis of ZFR and mH2A1 in THP-1 cells stably transduced with shRNAs specific to ZFR (shZFR) or non-targeting control shRNAs (shScr). Approximate MW based on size markers is shown. Graphs indicate mean ± SD; *$P < 0.05$, **$P < 0.01$ (two-tailed Student's $t$-test)

**ZFR regulates pre-mRNA splicing of many human genes**. As previous studies have implicated ZFR homologs in pre-mRNA splicing[14,15], we used MISO software to assess transcriptome-wide alternative splicing changes upon ZFR depletion from HEK-293TO cells[16]. These analyses revealed that ZFR knockdown caused widespread changes in splicing patterns, significantly affecting 276 alternative splicing events, as judged using previously established significance cutoffs (Fig. 2a; Supplementary Fig. 2A, Supplementary Data 2)[17]. The predominant form of alternative splicing detected was modulation of cassette exon inclusion (158 skipping/inclusion events). Among alternative splicing events affecting cassette exons, we observed a trend toward exon inclusion in the absence of ZFR, but exon-skipping events were also detected (Supplementary Fig. 2B).

Consistent with the previous finding that the *Drosophila melanogaster* ZFR homolog Zn72D is required for proper 5′ splice site (5′SS) selection in the *maleless* mRNA[14,15], we observed alternative 5′SS usage in 26 cases. We also observed alteration of 32 and 37 mutually exclusive splicing and intron retention events, respectively, while only 23 alternative 3′SS events were changed, less than expected based on the frequency of A3SS events in the data set (Supplementary Fig. 2A). Together, these data indicate that ZFR has an evolutionarily conserved role in alternative splicing, which can affect multiple types of splicing events.

**ZFR prevents aberrant splicing of mH2A1 mRNA**. In addition to elucidating a global role for ZFR in alternative splicing regulation, MISO analysis of our ZFR knockdown RNA-seq data revealed the mechanism by which ZFR promotes mH2A1 expression. Two major mH2A1 isoforms, designated mH2A1.1 and mH2A1.2, are produced by mutually exclusive use of exons 6b and 6a, respectively (Fig. 2b). Alternative splicing of these exons is regulated in a cell-type-specific manner, and the resulting proteins have distinct biochemical and regulatory properties[18–20]. The mH2A1.1 splice isoform retains residues required for interaction with poly-ADP ribose (PAR) and has been found to inhibit tumor progression and induced pluripotency, while mH2A1.2 lacks the ability to bind PAR and is widely expressed in cancer[18,21]. mH2A1.2 is the predominant isoform in HEK-293 cells, but MISO analysis and RT-PCR studies revealed efficient skipping of both exons 6a and 6b in ZFR knockdown cells (Fig. 2b,c). Importantly, skipping of both exons changes the translational reading frame of the resulting mRNA, introducing a premature termination codon in *mH2A1* exon 7 (Fig. 2c).

**Aberrantly spliced mH2A1 mRNAs are degraded by NMD**. mRNAs with PTCs located more than 55 nt upstream of the terminal exon–exon junction are expected to be efficiently degraded by NMD[22]. To determine whether ZFR depletion affected *mH2A1* mRNA stability, we pursued a metabolic labeling approach in which cells are treated with a short pulse of 5-ethynyl uridine (5-EU), a modified nucleoside that can be derivatized with biotin via click chemistry. Incorporation of 5-EU in newly synthesized RNAs allows specific purification and quantification of RNAs produced during the 5-EU incubation.

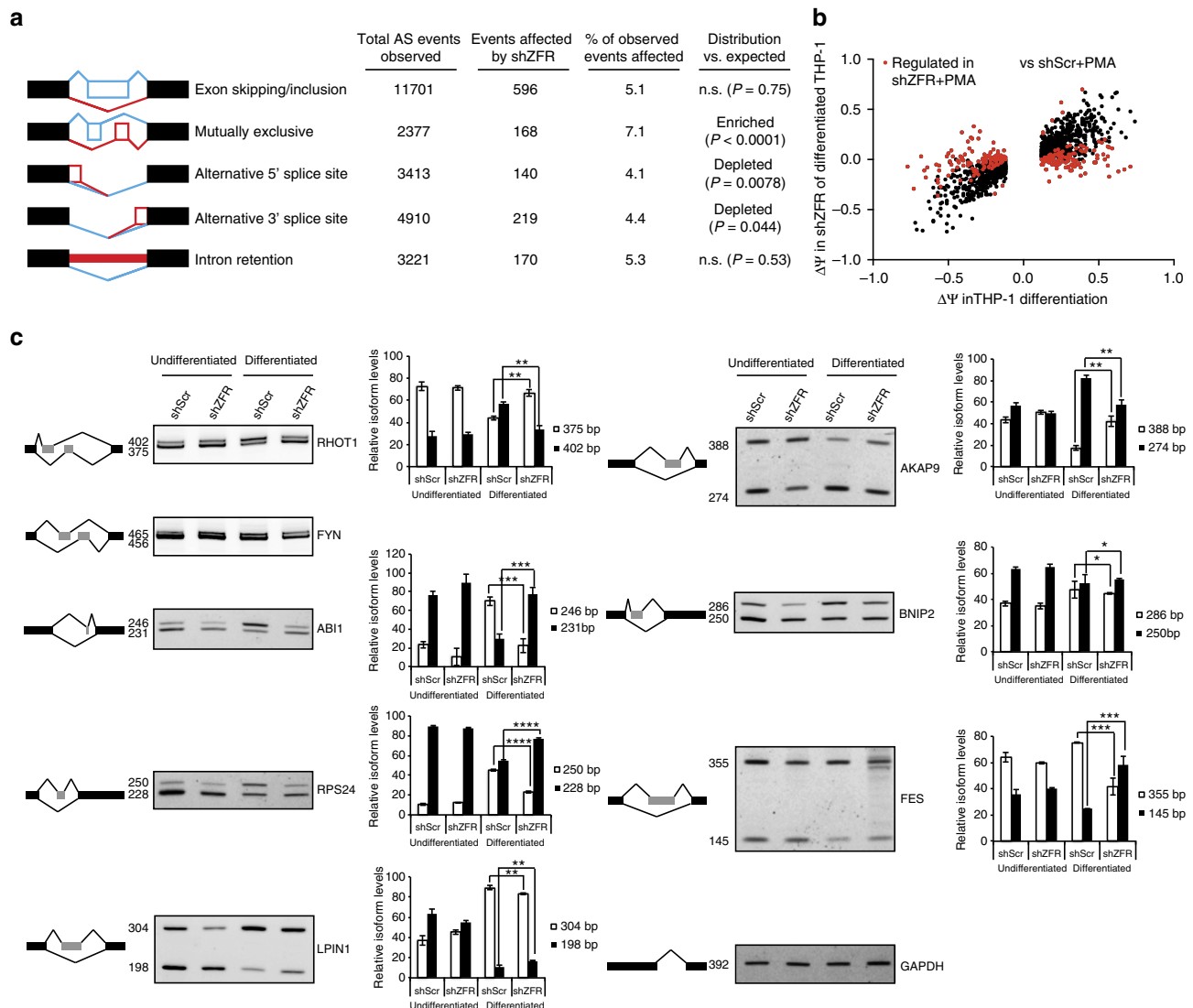

**Fig. 4** ZFR regulates alternative splicing in THP-1 differentiation. **a** Table of alternative splicing events affected by ZFR knockdown in differentiated THP-1 cells. Numbers of total alternative splicing events meeting read coverage cutoffs and events significantly changed by ZFR knockdown in MISO analysis (bayes factor > 20 and $|\Delta\Psi| > 0.12$) are indicated (see Methods for details). *P* values were derived from binomial tests of observed vs. expected frequencies. **b** Scatter plot of $\Delta\Psi$ values from comparisons of differentiated THP-1 cells treated with shScr and shZFR (*y*-axis) and $\Delta\Psi$ values from comparisons of differentiated shScr THP-1 cells and monocytic shScr THP-1 cells (*x*-axis). Events significantly changed (bayes factor > 20 and $|\Delta\Psi| > 0.12$) upon ZFR knockdown are indicated in red. **c** RT-PCR of the indicated transcripts in monocytic and macrophage-like THP-1 cells treated with ZFR-specific or control shRNAs, using primers flanking exons found to be regulated by MISO analysis. Usage of each isoform was calculated as percent of total ($n = 3$). Approximate MW based on size markers is shown. Graphs indicate mean ± SD; *$P < 0.05$, **$P < 0.01$, ***$P < 0.001$, ****$P < 0.0001$ (two-tailed Student's *t*-test). Co-migration of bands corresponding to mutually exclusive FYN isoforms precluded accurate quantification

Comparison of mRNA levels in the total and labeled pools of RNA can then be used to calculate rates of decay[23]. Consistent with the hypothesis that mis-spliced *mH2A1* mRNA is targeted by NMD, metabolic labeling and qRT-PCR showed that the *mH2A1* mRNA was significantly destabilized by ZFR knockdown (Fig. 2d). As an initial test of the NMD sensitivity of the aberrant *mH2A1* mRNA, we treated cells with cycloheximide, an inhibitor of translation and NMD. Indeed, cycloheximide treatment increased mH2A1 and canonical NMD target mRNA levels in ZFR knockdown cells without affecting *ZFR* mRNA expression[24] (Fig. 2e; Supplementary Fig. 2C, D). This increase was due to stabilization of the exon-skipped *mH2A1* isoform, as qRT-PCR with isoform-specific primers revealed a several-fold increase in the exon 5-exon 7 (E5/E7) product but no effect on exon 5-exon 6 (E5/E6) and exon 6-exon 7 (E6/E7) products in ZFR knockdown

cells following cycloheximide treatment (Fig. 2f). Further indicating that the shorter *mH2A1* isoform generated by ZFR depletion is degraded by NMD, knockdown of core NMD proteins UPF1 and SMG1 also increased expression of the E5/E7 isoform (Fig. 2g; Supplementary Fig. 2E, F).

**Distinct ZFR isoforms in monocytes and macrophages**. While characterizing a newly developed monoclonal antibody raised against the C-terminal DZF domain of ZFR, we screened several cell lines for ZFR expression. Despite the fact that ZFR is widely expressed in different tissues[25], full-length ZFR protein was only faintly detected in human monocytic THP-1 cell lysates, and smaller isoforms were instead recognized by the ZFR antibody (Fig. 3a). Interestingly, full length ZFR was strongly induced in

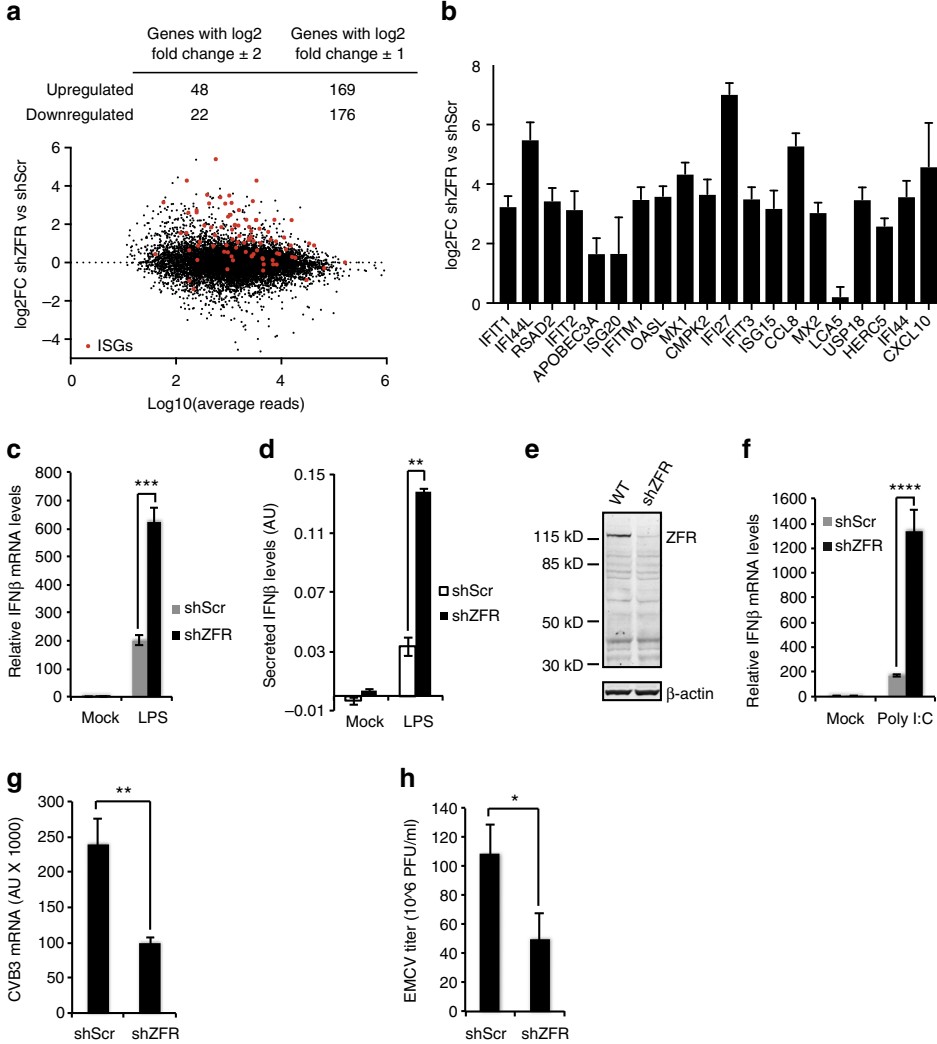

**Fig. 5** ZFR suppresses IFNβ induction and the antiviral response. **a** RNA-seq analysis of differentiated THP-1 cells stably transduced with shZFR or shScr ($n$ = 3). Top, numbers of genes upregulated or downregulated by ZFR depletion at the indicated thresholds. Bottom, scatter plot of fold change in mRNA abundance vs. number of reads derived from each gene, with mRNAs from previously identified ISGs shown in red[61,62]. **b** Induction of a previously characterized panel of ISGs[63] in differentiated THP-1 cells depleted of ZFR. Mean log2 fold change and SD from three RNA-seq replicates are shown. **c** qRT-PCR analysis of *IFNB1* in differentiated THP-1 cells stably transduced with shZFR or shScr and treated with LPS ($n$ = 3). **d** ELISA assay of secreted IFNβ in differentiated THP-1 cells stably transduced with shZFR or shScr and treated with LPS. **e** Immunoblot analysis of ZFR in HEK-293TO cells stably transduced with shZFR or shScr. β-Actin was used as a loading control. Approximate MW based on size markers is shown. **f** qRT-PCR analysis of *IFNB1* after poly(I:C) induction in HEK-293TO cells stably transduced with shZFR or shScr ($n$ = 4). **g** qRT-PCR analysis of Coxsackievirus B3 (CVB3) in the indicated HEK-293TO cells 6 h after infection ($n$ = 3). **h** Virus titer measured by plaque assay 48 h after EMCV infection in the indicated HEK-293TO cells ($n$ = 3). Graphs indicate mean ± SD; **$P$ < 0.01, ****$P$ < 0.0001 (two-tailed Student's $t$-test)

macrophage-like THP-1 cells differentiated with phorbol 12-myristate 13-acetate (PMA), with a corresponding decrease in the truncated ZFR isoform. This phenomenon was not isolated to THP-1 cells, as we observed a similar pattern of ZFR regulation upon differentiation of murine BMDMs and human primary CD14+ monocytes, albeit with cell-type-specific differences in mobility of truncated isoforms (Supplementary Fig. 3A and B).

To investigate the identity of this putative truncated ZFR isoform, we performed 5′ rapid amplification of cDNA ends (5′ RACE) on RNA from monocytic and macrophage-like THP-1 cells (Supplementary Fig. 3C). 5′RACE revealed a putative monocyte-enriched truncated isoform, which we validated by subsequent sequencing of 5′RACE products generated using multiple *ZFR*-specific primers. These analyses indicated that the short ZFR isoform was derived from an alternative transcription start site (TSS) in exon 13 (exon numbering from full-length

isoform; Supplementary Fig. 3D, E). Supporting this identification, publicly available CAGE-seq (Cap analysis gene expression) data identify a TSS cluster in this region that is preferentially used in monocytic cells (Supplementary Fig. 4A)[26]. Furthermore, ENCODE chromatin immunoprecipitation (ChIP) experiments show cell-type-specific enrichment of histone H3 lysine 27 acetylation, a marker of active promoters, in this region (Supplementary Fig. 3E)[27]. In mice, CAGE-seq data also indicate use of two clusters of alternative promoters predicted to generate ~65 kDa and ~15 kDa proteins, consistent with the major alternative protein isoforms we detected in undifferentiated mouse cells (Supplementary Fig. 3A and Supplementary Fig. 4B).

Transcripts produced from the alternative TSS clusters identified by CAGE-seq and 5′RACE are expected to produce proteins encoding the C-terminal DZF domain but lacking zinc fingers (Fig. 3a; Supplementary Fig. 3E). To further probe the

identities and possible functions of the truncated ZFR isoform, we cloned a cDNA containing the 5′UTR and truncated ZFR ORF present in the 5′RACE products in a C-terminally FLAG-tagged mammalian expression vector. Expression of this cDNA-generated protein isoforms with mobility closely matching those observed in undifferentiated THP-1 cells (compare Fig. 3a and Supplementary Fig. 4C). As expected based on our RNA-seq data, overexpression of the truncated ZFR isoform was unable to rescue *mH2A1* mRNA in cells depleted of endogenous ZFR. In contrast, overexpression of a full-length *ZFR* cDNA resulted in a 12-fold increase in *mH2A1* mRNA in ZFR-depleted cells, with no significant effect in control cells (Supplementary Fig. 4D). Together, these data suggest that monocytic cells preferentially use an alternative TSS to produce ZFR proteins deficient for splicing regulatory activity.

To determine whether the ZFR-mediated regulation of mH2A1 we initially identified in HEK-293TO cells is also operative in macrophages, we examined mH2A1 protein expression by immunoblotting of THP-1 and mouse BMDM cell extracts before and after differentiation (Fig. 3b). Importantly, the increase in full-length ZFR expression upon differentiation was mirrored by induction of mH2A1 expression in both cell types. Further, we used shRNA-mediated depletion of ZFR in macrophage-like THP-1 cells to confirm the role of ZFR in promoting mH2A1 expression in this cellular context. As in HEK-293TO cells, depletion of ZFR also reduced mH2A1 expression in THP-1 cells (Fig. 3c). Tight regulation of ZFR expression in macrophage differentiation suggests that ZFR may play an important post-transcriptional regulatory role in these cells, perhaps in part by modulating mH2A1 levels.

**ZFR regulates many splicing events in THP-1 differentiation**. We next investigated the function of ZFR in monocytic and macrophage-like THP-1 cells by stably depleting ZFR with shRNAs and performing RNA-seq of cells before and after differentiation with PMA. To aid identification of alternative splicing events by MISO, we generated ~100 million paired-end 100 bp reads per triplicate sample from ZFR knockdown and control monocytic and macrophage-like THP-1 cells. As expression of full-length ZFR was highest in macrophages (Fig. 3a; Supplementary Fig. 3A and B), we first compared isoform usage in differentiated ZFR-depleted and control cells. This approach enabled the identification of 1293 ZFR-dependent alternative splicing events in macrophage-like THP-1 cells (Fig. 4a; Supplementary Data 3). Notably, mutually exclusive exon usage was significantly enriched among events regulated by ZFR in differentiated THP-1 cells (168 events; $P < 0.0001$). Widespread regulation of exon skipping/inclusion and intron retention was also observed, while alternative 5′ and 3′ SS usage was regulated less frequently than expected based on the underlying distribution of events meeting read coverage cutoffs. These findings are broadly consistent with the pattern of alternative splicing observed by sequencing HEK-293TO cells at lower depth, but the increased frequency of ZFR-dependent mutually exclusive events in THP-1 cells suggests that its splicing regulatory function can be modulated in a cell-type-specific manner.

Due to our observation of differential ZFR isoform expression in monocytes and macrophages, we investigated the importance of ZFR in determining alternative splicing patterns upon THP-1 differentiation. Following PMA treatment of control THP-1 cells, we observed alteration of 1157 alternative splicing events in the normal process of PMA-induced THP-1 differentiation (Fig. 4b, ΔΨin shScr-PMA vs. shScr; Supplementary Data 3). Among the alternative splicing events that underwent significant changes in differentiation of control cells, approximately 20% were

attenuated by ZFR depletion (Fig. 4b, ΔΨin shZFR-PMA vs. shScr). Importantly, of the differentiation-dependent alternative splicing events that were significantly altered by ZFR depletion, almost all (230 of 239; Fig. 4b, events different between PMA-treated ZFR and control knockdown cells are highlighted in red) reverted to the splicing pattern observed in monocytic THP-1 cells. To corroborate the MISO analysis of ZFR-dependent splicing changes in THP-1 cells, we performed RT-PCR using primers flanking several putatively regulated exons (Fig. 4c). RT-PCR confirmed that differentiation-induced alternative splicing of several genes, including *FYN*, *FES*, *RHOT1*, *ABI1*, *LPIN1*, *AKAP9*, *BNIP2*, and *RPS24*, was impaired by ZFR depletion. In contrast to the widespread changes in alternative splicing in the absence of ZFR, we did not observe perturbation of THP-1 differentiation upon ZFR depletion, as expression levels of commonly used differentiation markers were similar between cells treated with non-silencing control shRNAs and ZFR shRNAs (Supplementary Fig. 5A)[28]. Furthermore, the global pattern of gene expression changes upon differentiation with PMA was highly correlated between ZFR knockdown and control cells (Supplementary Fig. 5B). Together, these data indicate that ZFR has an important role in regulation of macrophage-specific patterns of alternative splicing.

**ZFR depletion enhances type I interferon production**. In addition to changes in numerous alternative splicing events, we observed twofold or greater upregulation or downregulation of mRNAs from 169 and 176 genes, respectively, upon ZFR knockdown in macrophage-like THP-1 cells (Fig. 5a; Supplementary Data 4). To begin to determine if specific aspects of macrophage function were altered upon ZFR knockdown, we used Gene Set Enrichment Analysis. Interestingly, this analysis revealed that targets of type I interferon signaling were preferentially induced in differentiated THP-1 cells depleted of ZFR (known ISGs are highlighted in red in Fig. 5a, see also Supplementary Fig. 6A, Supplementary Data 5). Genes upregulated by ZFR depletion included many well-characterized ISGs, suggesting that ZFR suppresses basal type I interferon signaling in the absence of infection (Fig. 5a,b).

As our RNA-seq experiments indicated a role for ZFR in suppressing the type I interferon response, we probed a role for ZFR as a negative regulator of innate immune signaling.

Following differentiation of THP-1 cells stably depleted of ZFR into macrophage-like cells, we treated them with the TLR4 ligand lipopolysaccharide (LPS) to mimic bacterial infection.

Strikingly, ZFR depletion resulted in approximately fivefold higher induction of both IFNβ mRNA and protein in differentiated THP-1 cells by LPS treatment (Fig. 5c,d). In addition to studying the effects of TLR signaling in macrophages, we extended our studies to test whether stable ZFR knockdown in HEK-293 cells would also enhance the cellular response to double-stranded RNA. ZFR knockdown and control shRNA-expressing cells were transfected with the viral dsRNA mimic polyinosinic:polycytidylic acid (poly(I:C)), which is recognized by the RIG-I and MDA5 dsRNA sensors. Poly(I:C) treatment leads to rapid induction of *IFNB1* transcription and subsequent induction of ISGs. In ZFR knockdown cells (Fig. 5e), *IFNB1* induction was enhanced to levels approximately sixfold higher than those observed in control cells following poly(I:C) treatment (Fig. 5f), with a corresponding increase in multiple downstream ISGs (Supplementary Fig. 6B). Together, these data suggest that full-length ZFR is induced during macrophage development, where it acts to dampen IFNβ production in response to PAMP detection.

Further, the hyperactivation of IFNβ in response to PAMPs in ZFR-depleted cells indicated a role for ZFR in determining the

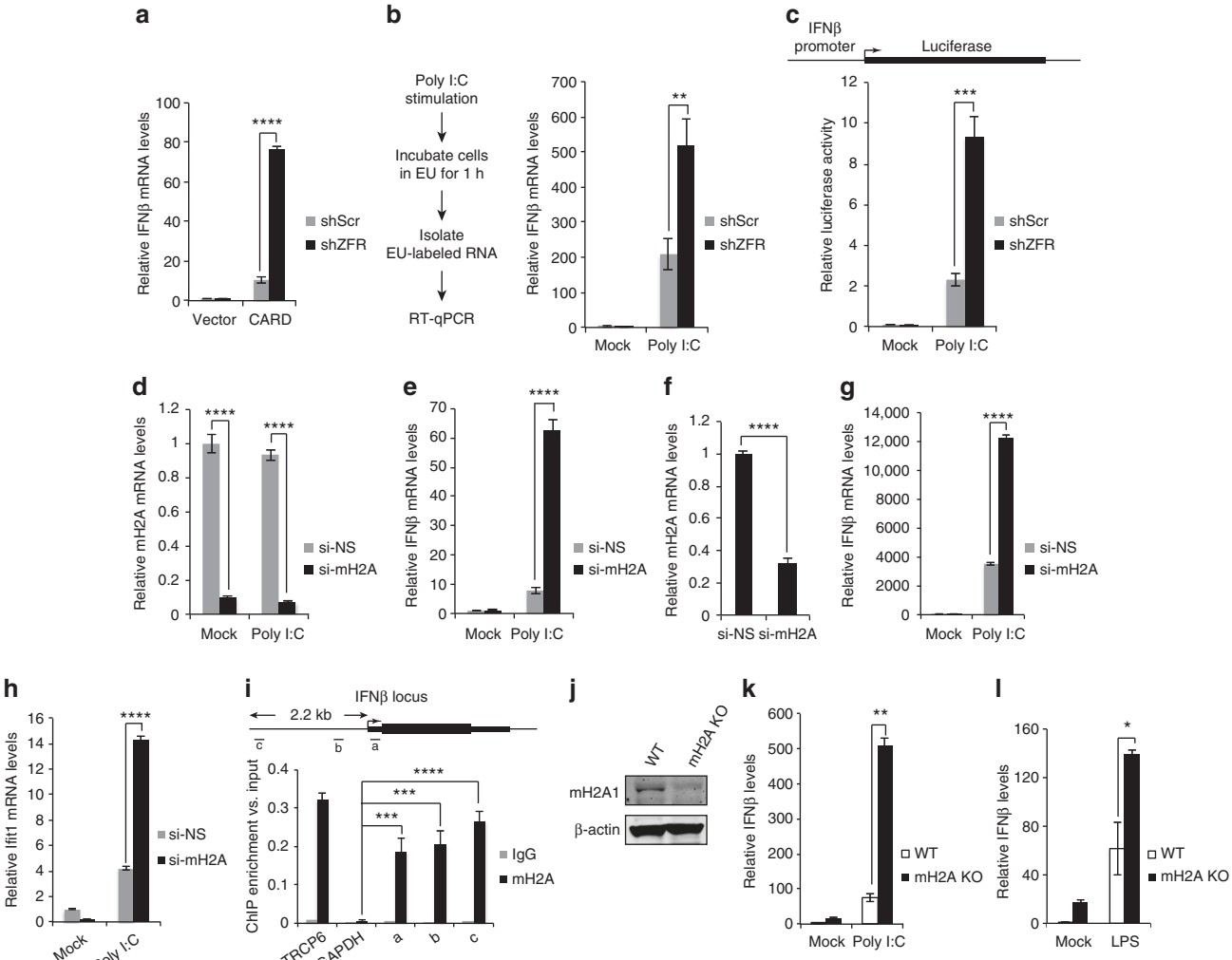

**Fig. 6** mH2A1 is responsible for ZFR-mediated IFNβ repression. **a** qRT-PCR analysis of *IFNB1* after transient transfection of a plasmid DNA-expressing 3xFLAG-tagged RIG-I CARD domain in HEK-293TO cells stably transduced with shZFR or shScr (n = 3). **b** Schematic of experiment to analyze synthesis of *IFNB* mRNA after poly(I:C) stimulation (left). qRT-PCR of newly synthesized *IFNB1* mRNA 24 h after poly(I:C) stimulation (right; n = 3). **c** Dual luciferase assay of cells transfected with plasmids expressing firefly luciferase under the control of the *IFNB1* promoter, with and without poly(I:C) stimulation (n = 3). A plasmid constitutively expressing Renilla luciferase was used as control. **d** qRT-PCR analysis of mH2A1 in HEK-293TO cells transfected with si-mH2A1 or si-NS with and without stimulation with poly(I:C) (n = 3). **e** qRT-PCR analysis of *IFNB1* as in D (n = 3). **f** qRT-PCR analysis of *mH2A1* in mouse BMDMs transfected with si-mH2A1 or si-NS (n = 3). **g** qRT-PCR analysis of *Ifnb1* in BMDMs transfected with the indicated siRNAs and stimulated with poly(I:C) (n = 3). **h** qRT-PCR analysis of *Ifit1* as in g (n = 3). **i** Chromatin immunoprecipitation-qPCR (ChIP-qPCR) analysis of mH2A1 binding to the *IFNB1* promoter region in HEK-293TO cells. Isotype-matched IgG was used as a control. Fold enrichment over input is shown for each location (n = 3). **j** Immunoblot analysis of mH2A1 in BMDMs from WT or macroH2A double knockout mouse. Approximate MW based on size markers is shown. **k** qRT-PCR analysis of *Ifnb1* in BMDMs stimulated with poly(I:C) or (**l**) LPS. Graphs indicate mean ± SD; *P < 0.05, **P < 0.01, ***P < 0.001, ****P < 0.0001 (two-tailed Student's t-test)

ability of cells to combat viral infection. To test this hypothesis, we challenged control and ZFR-depleted HEK-293 cells with two RNA viruses that trigger type I interferon responses, Coxsackievirus B3 (CVB3) and encephalomyocarditis virus (EMCV)[29,30]. Consistent with its effects on IFNβ induction, ZFR depletion caused a marked decrease in replication of both viruses (CVB3, Fig. 5g; EMCV, Fig. 5h). Thus, we have identified ZFR as a strong inhibitor of type I interferon induction and the cellular response to viral infection.

**ZFR inhibits IFNβ transcription.** The ability of ZFR to regulate IFNβ induction in response to both dsRNA and LPS suggests that it affects steps downstream of PAMP sensing shared by RIG-I/MDA5-dependent and TLR-dependent pathways. To further investigate whether the effect of ZFR on IFNβ induction is independent of the mechanism of PAMP detection, we transiently

transfected HEK-293TO cells with a plasmid expressing the RIG-I CARD domain, which constitutively activates *IFNB1* transcription by binding to the MAVS adaptor protein[31]. Mimicking the enhanced response to dsRNA and LPS described above, cells depleted of ZFR showed elevated *IFNB1* mRNA levels following RIG-I CARD overexpression (Fig. 6a; Supplementary Fig. 7A). These data indicate that ZFR acts after the initial steps of pathogen detection, in a manner independent of the specific sensing pathway activated.

Activation of IFNβ is regulated predominantly at the level of transcription in response to infection. Following PAMP sensing, signaling cascades mediate the activation of transcription factors including IRF3/7 and NF-κB, which bind the *IFNB1* promoter to stimulate transcription.

To investigate whether enhanced transcription was responsible for the large increase in *IFNB1* mRNA upon ZFR depletion, we

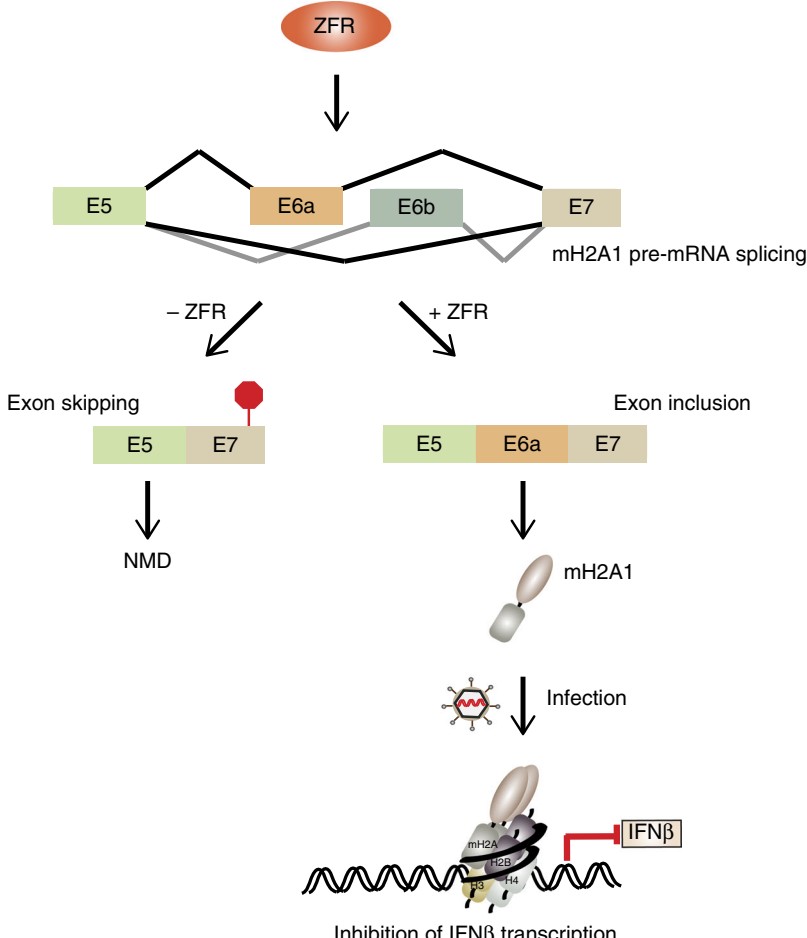

**Fig. 7** Model for ZFR-mediated regulation of IFNβ. ZFR is required for productive splicing of *mH2A1*. In the absence of ZFR, *mH2A1* pre-mRNA is mis-spliced and resulting transcript is degraded by NMD. In the presence of ZFR, mH2A1 protein is expressed and directly represses the *IFNB1* promoter

used metabolic labeling to measure mRNA synthesis following poly(I:C) treatment[23] (Fig. 6b). Cells were transfected with poly(I:C) and 24 h later incubated with 5-EU, allowing assessment of mRNA production during the labeling interval. Quantitative RT-PCR assays showed higher levels of labeled *IFNB1* mRNA in ZFR-depleted cells than control cells, indicating increased *IFNB1* transcription in the absence of ZFR (Fig. 6b).

The metabolic labeling data presented above suggest a role for ZFR in regulating *IFNB1* transcription, but could also be explained by rapid turnover of *IFNB1* mRNAs in the absence of ZFR. To distinguish between these two possibilities, we assayed the effect of ZFR depletion on expression of firefly luciferase driven by a minimal *IFNB1* promoter. In these experiments, firefly luciferase expression was also potently enhanced following poly(I:C) treatment of ZFR knockdown cells (Fig. 6c), consistent with a model in which ZFR-mediated repression targets the core *IFNB1* promoter.

Despite increased *IFNB1* transcription in the absence of ZFR, luciferase reporters containing multiple-binding sites for IRF3/IRF7 or NF-κB, the major transcription factors responsible for *IFNB1* induction, did not show enhanced activity in ZFR knockdown cells (Supplementary Fig. 7B, C). Moreover, no change in the levels of IRF3 phosphorylation, a hallmark of IRF3 activation, were observed in ZFR knockdown and control cells after poly(I:C) induction (Supplementary Fig. 7D). These findings suggest that ZFR acts as a repressor of *IFNβ* transcription but does not disrupt IRF3/7 or NF-κB activity per se.

**ZFR-dependent IFNβ repression is mediated by mH2A1.** Multiple findings suggest that mH2A1 can either repress or activate inducible genes in a context-dependent manner[32–34], positioning mH2A1 as a candidate for ZFR-dependent inhibitor of *IFNB1* transcription. To directly test the role of mH2A1 in *IFNB1* transcription, we depleted mH2A1 with specific siRNAs and treated cells with poly(I:C) (Fig. 6d). Loss of mH2A1 precisely phenocopied the effect of ZFR knockdown in HEK-293TO cells, as several-fold higher induction of *IFNB1* was observed in mH2A1-depleted cells than control cells (Fig. 6e; Supplementary Fig. 7E, F). Furthermore, knockdown of mH2A1 in primary mouse BMDMs also led to increased induction of *Ifnb1* and downstream ISG *Ifit1* in response to poly(I:C) (Fig. 6f–h). To determine whether mH2A1 directly mediates repression of *IFNB1* transcription, we performed ChIP using antibodies against endogenous mH2A1 protein.

ChIP-qPCR showed enrichment of mH2A1 throughout the *IFNB1* promoter, similar to that observed at previously characterized mH2A1 target genes[21] (Fig. 6i). Moreover, our ChIP results are consistent with genome-wide ChIP-Seq experiments identifying enrichment of mH2A1 at the *IFNB1* promoter[34]. Similar to previous studies investigating the role of mH2A1 in regulation of inducible genes, we did not observe changes in mH2A1 occupancy of the *IFNB1* locus following poly(I:C) stimulation (Supplementary Fig. 7G).

To further probe the physiological role of macroH2A in innate immune responses, we took advantage of previously described

mice lacking both macroH2A1 and macroH2A2 paralogs (*H2afy/H2afy2*; Fig. 6j)[35]. BMDMs from mH2A double knockout mice exhibited moderately elevated *Ifnb1* mRNA under basal conditions, and showed greatly enhanced induction of *Ifnb1* mRNA at early time points following treatment with either poly (I:C) or LPS, relative to wild-type mice (Fig. 6k,l). Hyperactivation of *IFNB1* production was accompanied by subsequent increases in levels of ISG mRNAs (Supplementary Fig. 7H–J), indicative of a concerted upregulation of innate immune signaling in the absence of mH2A. In sum, our data suggest that the effect of ZFR on *IFNB1* transcription is achieved through direct binding and repression of the *IFNB1* promoter by mH2A1 in diverse mouse and human cell types, including macrophages.

## Discussion

We show here that ZFR is the linchpin of a mechanism that controls RNA processing and decay to strongly affect the magnitude of *IFNB1* transcriptional induction in response to infection (Figure 7). We find that ZFR depletion causes constitutive induction of many ISGs in differentiated THP-1 cells that have not been exposed to PAMPs and fivefold hyperactivation of IFNβ in THP-1 and HEK-293TO cells treated with LPS or poly(I:C), respectively. We show that aberrant *IFNB1* transcription in PAMP-exposed cells arises from the lack of mH2A1 in ZFR-depleted cells. However, mH2A1 has also been shown to bind and repress ISG promoters, raising the possibility that the constitutive ISG induction in ZFR-depleted cells could be partially due to loss of direct action of mH2A1 on these genes[33,34]. Taking the available data together, mH2A1 appears to both suppress the initiation of type I interferon signaling by inhibiting *IFNB1* transcription and by modulating the consequences of *IFNB1* transcription by directly repressing ISG promoters.

Control of mH2A1 by ZFR thus represents an unanticipated pathway through which cells modulate innate immune responses. Because of the highly deleterious consequences of type I interferon hyperactivation, numerous mechanisms exist to constrain their production and activities[36]. Negative regulation of interferon signaling takes place at multiple steps in the pathway but the strength and central importance of *IFNB1* transcription make it a particularly important regulatory target. Underlining the significance of such regulatory mechanisms, mutations in several genes responsible for either positively or negatively regulating IFN signaling have been found to cause human diseases collectively known as the type I interferonopathies[37]. These disorders are associated with continuous type I interferon production, which drives expression of proinflammatory cytokines and ISGs and results in autoimmunity, embryotoxicity, neurological phenotypes, skin disorders, immunosuppression, and other pathologies.

ZFR-mediated regulation of mH2A1 occurs through a surprising mechanism, as ZFR depletion causes efficient skipping of the normally mutually exclusive mH2A1.1-specific and 1.2-specific exons, after which the mis-spliced product is degraded by NMD. Our findings suggest that this mechanism is used in macrophage differentiation to control mH2A1 expression and to dampen IFNβ expression. Due to the breadth of mH2A1 functions, we expect that ZFR-mediated regulation of mH2A1 expression is important in many biological contexts. Intriguingly, depletion of ZFR has recently been reported to inhibit pancreatic cancer cell growth, and the well-established significance of mH2A1 in cancer suggests that this effect could be linked to altered expression of the histone variant[38]. Further, mH2A1.1 is induced upon differentiation of human embryonic stem cells, after which it helps to maintain the differentiated state[39–41],

raising the possibility that ZFR is involved in regulation of cellular differentiation and reprogramming through mH2A1.

ZFR's effect on IFNβ expression through mH2A1 is just one aspect of its role in alternative splicing regulation. We present the first genome-wide-analyses of ZFR functions, showing that it has major impacts on splicing patterns in distinct cell types. The propensity of ZFR to modulate mutually exclusive exons in THP-1 cells is of particular interest, as future ZFR studies may shed light on the mechanisms underlying this poorly understood class of alternative splicing events[42]. Our findings suggest that ZFR's splicing regulatory activities may be modulated in a cell-type-specific manner, either by concurrent action of additional proteins on shared mRNA targets or by direct physical interactions.

We initially chose to probe the role of ZFR in macrophages due to our observation that full-length ZFR expression is induced in macrophages. 5′RACE experiments suggest that the truncated protein is encoded by a previously annotated mRNA arising from an alternative TSS; however, we cannot rule out other possible mechanisms to generate alternative ZFR isoforms in a cell-type-specific manner. Supporting this identification, the truncated isoform is recognized by the monoclonal anti-ZFR antibody, which was raised against the C-terminal DZF domain. Our data indicate that this isoform lacks the ability to affect mH2A1 splicing, presumably because ZFR relies on its zinc fingers for RNA recognition (Supplementary Fig. 4D). However, it remains to be determined whether the isolated ZFR DZF domain can affect certain splicing events by interacting with proteins such as ILF2 and/or ILF3[13]. We note that ZFR depletion in monocytic THP-1 cells does affect some splicing events, which could either be due to loss of the small amount of full-length ZFR expressed in these cells or to perturbation of the truncated isoform (Fig. 4c; Supplementary Data 3).

In line with tight regulation of ZFR expression in macrophage differentiation, we show that ZFR has an important role in regulating differentiation-dependent alternative splicing events. A large subset (~20%) of the alternative splicing changes we observed upon differentiation of control THP-1 cells were attenuated in cells depleted of ZFR. Among the events identified by MISO analysis, we confirmed several events affecting proteins previously determined to have important functions in macrophages. For example, ABI1 is important for actin remodeling and phagosomal acidification[43,44], while LPIN1 promotes proinflammatory signaling downstream of TLR4[45]. We also found that ZFR regulates well-characterized mutually exclusive exons in the *FYN* receptor tyrosine kinase gene that yield protein isoforms with differential kinase activities, termed FynT and FynB[46,47]. We find that the FynT isoform is induced in macrophages in a ZFR-dependent manner, likely enhancing FYN kinase activity in macrophage-like cells. Interestingly, ZFR also regulates a FYN target, the FES kinase, in the course of THP-1 differentiation. FYN phosphorylation of FES has been reported to enhance FES activity by relieving autoinhibition[48]. In turn, FES can regulate innate immune signaling by phosphorylating STAT3 and promoting TLR4 internalization[49,50]. ZFR promotes inclusion of a *FES* exon encoding an SH2 domain important for full FES kinase activity[51], potentially balancing the effects of increased FynT production. Together, these alternative splicing events represent several additional pathways by which ZFR can modulate the response to infection.

## Methods

**Cell culture**. HEK-293 Tet-Off (HEK-293TO) cells (Clontech) were maintained in Dulbecco's Modified Eagle's Medium (DMEM; Thermo Scientific) supplemented with 10% FBS (Gibco), 100 U/mL of penicillin, 100 µg/mL streptomycin (Gibco), and 0.3 mg/mL of L-glutamine (Gibco), at 37 °C and 5% CO₂. Human monocytic THP-1 cells (ATCC) were maintained in ATCC-formulated RPMI-1640 medium supplemented with 10% FBS (Gibco), 0.05 mM 2-mercaptoethanol (Sigma), and 100 U/mL of Penicillin, 100 µg/mL Streptomycin (Gibco). Cell lines were obtained directly from

ATCC or Clontech as indicated and were periodically tested for mycoplasma contamination using the MycoAlert Mycoplasma Detection Kit (Lonza).

**Cell treatments.** Low molecular weight poly(I:C) and *Escherichia coli* LPS were purchased from Invivogen and Sigma, respectively. For induction with poly(I:C), HEK-293TO cells were plated at $2.5 \times 10^5$ cells/mL in 24-well plates and transfected with 1 µg/mL poly(I:C) using TurboFect (Thermo Scientific); 2 µL of TurboFect was used per µg of poly(I:C). After 24 h, cells were washed with PBS and harvested for RNA using the MagMAX-96 RNA Isolation kit (Thermo Scientific). THP-1 cells were differentiated into macrophage-like cells with 5 ng/mL of PMA (Sigma) for 72 h. Cells were washed with PBS and induced with media containing LPS (1 ng/mL). Cycloheximide was purchased from Sigma, and $2.5 \times 10^5$ HEK-293TO cells were treated with 50 µg/mL cycloheximide for 4 h. Cells were washed with PBS and harvested for RNA as above. Differentiated Mouse BMDMs were treated with LPS (100 ng/mL) or poly(I:C) (5 µg/mL) for immunostimulation.

**BMDM isolation.** The macroH2A double-KO mice were kindly provided by Dr. John Pehrson. (University of Pennsylvania)[35]. BMDMs were prepared from age and size-matched 129/S6 background mice without blinding or randomization. The procedures used in this work were approved by the NICHD animal care and use committee with ASP #17-044. Bone marrow cells were cultured in DMEM/F12 medium containing 10% FBS and 20% L929 conditioned medium as a source of M-CSF for differentiation[52].

**Plasmids.** Luciferase reporter constructs containing interferon-sensitive response elements (ISRE, pGL4.45) or NF-κB response elements (pGL4.32) were purchased from Promega. To construct the *IFNB1* promoter reporter plasmid, the region 125 bp upstream of the transcriptional start site of the human *IFNB1* gene was PCR amplified from HEK-293T genomic DNA and inserted into the SacI and XhoI sites of pGL4.45 vector (Promega), replacing the ISRE sequences. To construct the RIG-I-CARD expression plasmid, an N-terminal 687 bp fragment of the RIG-I coding sequence was PCR amplified using Phusion High-Fidelity DNA polymerase (NEB) and cloned into the HindIII and BamHI sites of pcDNA5 (Thermo Scientific), previously modified in-house to encode an N-terminal 3XFLAG tag. For cloning open reading frames, cDNA was synthesized using 0.5 µg of total RNA from HEK-293T cells and SuperScript III (Thermo Scientific) using an oligo(dT) primer. Full-length ZFR cDNAs were PCR amplified and cloned into the tetracycline-inducible expression vector pcDNA5/FRT/TO (Thermo Scientific), which was also modified to harbor an N-terminal 3XFLAG tag. A plasmid expressing siRNA/shRNA-resistant full-length ZFR was generated by introducing synonymous mutations in pcDNA/FRT/TO/NX/3XFLAG-ZFR using the QuickChange II Site-directed Mutagenesis Kit (Agilent) using oligonucleotides sh-Resistant-ZFR-F and sh-Resistant-ZFR-R (Supplementary Data 6).

**RNAi.** shRNA-mediated silencing of ZFR was achieved in HEK-293TO cells using the miR-30-based shRNA lentiviral vector pGIPZ (Dharmacon, GE Healthcare). DNA oligos encoding shRNAs specific to ZFR (shZFR, 5′-TGTGTAGGAGACA-TAAGCA-3′) or a non-targeting control sequence (shScr, 5′-TCTCGCTTGGGCGAGAGTAAG-3′) were inserted into pGIPZ. To prepare viral particles, $3.8 \times 10^6$ HEK-293T cells were co-transfected with pGIPZ (10 µg), along with 7.5 µg pMDG[53] (VSV-G envelope) and 2.5 µg p8.9[53] (HIV-1 gag-pol) in 10 cm culture plates using 40 µL of TurboFect transfection reagent (Thermo Scientific). Viral supernatants were harvested after 72 h. HEK-293TO ($5 \times 10^5$) were transduced with 1 mL of viral supernatant in the presence of polybrene (8 µg/mL) in 6-well dishes. Media was replaced after 16 h, and cells were selected with puromycin (2.5 µg/mL) beginning at least 48 h after transfection. Depletion of ZFR was measured after 5 days by qRT-PCR or western blot analysis, and cells were maintained in the presence of puromycin. Stable depletion of ZFR was measured periodically by qRT-PCR. For transduction of human monocytic THP-1 cells, a pLKO.1 vector containing a shRNA targeting the human ZFR coding sequence (TRCN0000016966: 5′-CCGGGCCAAGGTGCAACTCAGTATACTCGAGTATA CTGAGTTGCACCTTGGCTTTTT-3′) was obtained from Sigma-Aldrich, and virus particles were prepared as described above. For transduction, $1 \times 10^6$ cells were mixed with 100 µL of 50-fold concentrated virus in the presence of polybrene (8 µg/mL) and spun at 1200 rpm for 1 h at 28 °C. Virus was replaced with fresh media, and cells were selected with puromycin (5 µg/mL) after 48 h. HEK-293TO cells ($3.75 \times 10^4$) were reverse-transfected with 30 pmol of siRNAs[54] (Silencer Select, Thermo Scientific) in 24-well plates using Lipofectamine RNAiMAX (Life Technologies). For RNA-Seq studies, $1.5 \times 10^5$ cells were transfected with 120 pmol of siRNAs in 6-well plates. Seventy-two hours after transfection, RNA was isolated using the RNeasy Mini kit (Qiagen). Differentiated mouse BMDMs were reverse-transfected in 24-well plates with 50 pmol of siRNA using Lipofectamine RNAi-MAX (Life Technologies). HEK-293TO cells were reverse-transfected with the following siRNAs: (si-ZFR: 5′-GCACUUAAAAGGGCGAAGATT-3′, si-mH2A: 5′-GCUAAAAGGAGUCAC CAUATT-3′, si-mH2A-2: 5′-CGGUGUACUUCG UGCUUUUUT-3′, si-UPF1: 5′-GAUGCAGUUCCGCUCCAUUTT-3′[55], si-SMG1: 5′-UGG AAUAGGUCUAACAGCATT-3′, si-NS: 5′-UAAGGCUAUGAAGAGA UAC-3′). Differentiated mouse BMDMs were reverse-transfected with mouse si-mH2A1: 5′-GC AUGCUUCGGGUACAUCAATT-3′.

**Detection of RNA.** Total RNA was isolated with the RNeasy Mini Kit (Qiagen) with on-column DNase digestion (Qiagen), or with MagMAX RNA isolation kit (Thermo Scientific). RNA concentration and quality was measured on a NanoDrop 1000 spectrophotometer (Thermo Scientific). cDNA was synthesized with Maxima First Strand cDNA synthesis kit (Thermo Scientific), diluted 1:20 with water, and used for qPCR using SYBR Green qPCR master mix (FastStart Essential DNA Green Master Mix, Roche) on a LightCycler 96 thermocycler (Roche). Relative levels of RNAs were calculated by the ddCt method using GAPDH or UBC as internal controls. See Supplementary Data 6 for sequences. Average and standard deviation values are from at least three biological replicates.

Semi-quantitative RT-PCR was performed with Phusion polymerase (NEB), run on 6–20% non-denaturing polyacrylamide gels, stained with SYBR Gold (ThermoFisher scientific), and analyzed on a ChemiDoc XRS system (BIO RAD). See Supplementary Fig. 9 for uncropped gels.

**Immunoblot analysis.** Cells were lysed in RIPA buffer (10 mM Tris-HCl pH 8.0, 1 mM EDTA, 0.5 mM EGTA, 150 mM NaCl, 1% NP40, 0.1% sodium deoxycholate, 0.1% SDS, and Halt Protease Inhibitor cocktail (Thermo Scientific)) for 1 h at 4 °C, and centrifuged at $9000 \times g$ for 10 min. Protein concentration was measured using the Pierce 660 nm Protein Assay Reagent (Thermo Scientific) on an Infinite M200 PRO multimode plate reader (Tecan). Equal amounts of protein (25 µg per lane) were loaded on a 4–12% Nupage Novex Bis-Tris protein gel (Thermo Scientific), transferred to 0.45 µM Nitrocellulose membrane (Bio-Rad) with an XCell II Blot Module (Thermo Scientific). Membranes were blocked overnight with Blocking Buffer for Fluorescent Western Blotting (Rockland), and blotted with primary antibodies against ZFR (custom mouse monoclonal, see below; 1:500), mH2A (Abcam, ab37264, 1:1000), IRF3 (Cell Signaling, 4302, 1:1000), phospho-IRF3 (Cell Signaling, 11904, 1:2000). For loading controls, antibodies against β-actin (Cell Signaling, 3700, 1:1000), PABPC1 (Abcam, ab6125, 1:1000), or GAPDH (Santa Cruz, sc-365062, 1:10,000) were used. Fluorescently labeled secondary antibodies (Thermo Scientific) were used at 1:10,000, and membranes were analyzed on an Odyssey Imaging System (LI-COR). See Supplementary Fig. 8 for uncropped images. Lysates from primary monocytes isolated and differentiated into macrophages as previously described were provided by Ying-Han Chen and Nihal Altan-Bonnet[56].

**Luciferase assays.** A total of $1 \times 10^5$ cells were co-transfected with 100 ng of Firefly luciferase reporter plasmid and 100 ng of plasmid expressing *Renilla* luciferase as a control. A volume of 1 µL of TurboFect transfection reagent (Thermo Scientific) per µg of DNA was used for transfection throughout the experiments. After 48 h of transfection, cells were stimulated with poly(I:C) (5 µg/mL) for 24 h and lysed to measure luciferase activity using the Dual Luciferase Reporter Assay kit (Promega). Measurements from Firefly luciferase activity were normalized to that of *Renilla* luciferase.

**Anti-ZFR monoclonal antibody generation.** Custom monoclonal antibodies were raised against amino acids 703–1074 of the human ZFR protein (GenScript).

**RNA-Seq.** For RNA-seq studies, RNA integrity was checked on an Agilent Bioanalyzer 2100, and 1 µg of total RNA was used per sample for library preparation. Ribosomal RNA was depleted from total RNA with Ribo-Zero rRNA Removal Kits (Epicentre), and purified RNA was then converted to cDNA with the Illumina TruSeq Stranded Total RNA Sample Preparation Kit (Illumina). cDNA libraries were run on an Illumina MiSeq instrument for quality assessment, and successful libraries were sequenced on the Illumina HiSeq 2000 or 3000 platform. Sequencing reads in FASTQ format were aligned to the reference genome hg19 using STAR aligner[57]. Differential gene expression analysis was performed using the DESeq2 package[58], and alternative splicing analysis was performed on pooled STAR alignments using MISO, with hg19 version 2 annotations[16]. Events were expression-filtered to require at least one read from each isoform and at least 10 total reads, and significance was determined using the previously described criteria of bayes factor > 20 and $|\Delta\Psi| > 0.12$[17]. For event tabulation, duplicate events sharing one or more splice site coordinates were removed. Functional classification of differentially expressed genes was performed using Gene Set Enrichment Analysis software[59].

**CVB3 infection.** A total of $1.5 \times 10^5$ HEK-293TO cells were infected with CVB3 (MOI = 10) for 6 h at 37 °C. Cells were washed with PBS, and total RNA was isolated using the RNeasy Mini Kit (Qiagen). Virus accumulation was analyzed by qRT-PCR using CVB3-specific primers (Supplementary Data 6).

**EMCV infection.** HEK-293TO cells were infected with EMCV (MOI = 1) in serum-free and antibiotic-free medium for 1 h at 37 °C, washed, and allowed to proceed in culture for 48 h with DMEM medium containing 10% FBS and antibiotics. Cells were fixed with methanol and incubated with amido black solution (Sigma) for 20 min at room temperature. Plaque assays were performed with L929 cells incubated with viral supernatants, followed by incubation with agar overlays

for 24 h at 37 °C Plaques were visualized by staining with 0.02% neutral red. Viral titers were expressed as plaque-forming units.

**Isolation of newly synthesized RNA**. A total of $2.5 \times 10^5$ HEK-293TO cells were grown in the presence of 0.2 mM 5-EU for 1 h to incorporate 5-EU in newly synthesized RNA. Total RNA was isolated using the RNeasy Mini kit (Qiagen), and 2.5 μg RNA was biotinylated with 0.5 mM biotin-azide using the Click-iT Nascent RNA Capture Kit (Thermo Scientific) and ethanol precipitated. An aliquot of 1 μg of biotinylated RNA was heated at 70 °C for 5 min, mixed with 50 μL of MyOne Streptavidin T1 magnetic beads, and incubated at RT for 30 min with shaking at 1200 rpm. Beads were washed on a magnet five times with 500 μL of Click-iT reaction wash buffer 1, and five times with Click-iT reaction wash buffer 2. Finally, cDNA was synthesized on beads using the Maxima First Strand cDNA synthesis kit (Thermo Scientific) and used for qPCR as described above.

**RIG-I CARD overexpression**. A total of $5 \times 10^4$ cells in 24-well plates were transfected with 100 ng of pcDNA5-3XFLAG-CARD or empty pcDNA5-3XFLAG plasmid using TurboFect transfection reagent (Thermo Scientific). Seventy-two hours after transfection, cells were harvested for protein and RNA analysis.

**Chromatin immunoprecipitation**. A total of $1 \times 10^7$ cells were crosslinked with 0.8% formaldehyde for with rotation at room temperature for 10 min and subsequently quenched with 0.125 M glycine[60]. After washing twice with ice-cold PBS, cells were resuspended in 1 mL of buffer containing 50 mM HEPES pH 7.5, 140 mM NaCl, 1 mM EDTA, 10% glycerol, 0.5% NP-40, 0.25% Triton X-100, and Halt protease inhibitor cocktail (Thermo Scientific), and kept on ice for 10 min. Nuclei were pelleted at $1350 \times g$ for 5 min at 4 °C and washed with 1 mL of buffer containing 10 mM Tris-HCl pH 8, 200 mM NaCl, 1 mM EDTA, 0.5 mM EGTA, and Halt protease inhibitor cocktail (Thermo Scientific) and kept on ice for 10 min. Nuclei were pelleted at $1350 \times g$ for 5 min and resuspended in lysis buffer containing 10 mM Tris-HCl pH 8, 100 mM NaCl, 1 mM EDTA, 0.5 mM EGTA, 0.1% Na-deoxycholate, 0.5% Na-lauryl sarcosine, and Halt Protease Inhibitor Cocktail (Thermo Scientific). Nuclei were lysed by passing through a 25-gauge needle four times before shearing chromatin with an ultrasonicator with built-in cooling device (Biorupter, Diagenode) for three sets of 10 cycles of 30 s each on the high setting with a 5-min interval between each set. Triton X-100 was added to 1%, and samples were spun at $15,000 \times g$ for 10 min at 4 °C. An aliquot of chromatin was reverse crosslinked (see below), and fragmentation of DNA to 100–400 bp was confirmed by agarose gel electrophoresis. Chromatin (25 μg) was diluted to 500 μL with buffer containing 10 mM Tris-HCl pH 8, 100 mM NaCl, 1 mM EDTA, 0.5 mM EGTA, 0.1% Na-deoxycholate, 0.5% Na-lauryl sarcosine, 1% Triton X-100, and Halt protease inhibitor cocktail (Thermo Scientific). An aliquot of 25 μL (5%) of chromatin was saved as input and the remainder was immunoprecipitated with 10 μg of macroH2A antibody (Abcam, ab37264) or normal rabbit IgG (Cell Signaling, 2729) overnight at 4 °C. A volume of 30 μL of Protein A/G magnetic beads (Thermo Scientific) was added and rotated at 4 °C for 1 h. Beads were washed sequentially for 5 min each with 500 μL of ice-cold wash buffer 1 (20 mM Tris-HCl pH 7.5, 150 mM NaCl, 2 mM EDTA, 1% Triton X-100, and 0.1% SDS), wash buffer 2 (20 mM Tris-HCl pH 7.5, 500 mM NaCl, 2 mM EDTA, 1% Triton X-100, and 0.1% SDS), wash buffer 3 (20 mM Tris-HCl pH 7.5, 250 mM LiCl, 2 mM EDTA, 1% NP-40, and 1% Na-deoxycholate), and TE buffer (10 mM Tris-HCl pH 8, 1 mM EDTA). Immunoprecipitated complexes were released from beads by adding 120 μL of elution buffer containing 20 mM Tris-HCl pH 8, 10 mM EDTA, 1% SDS, and 50 mM NaHCO₃, mixed at 1200 rpm for 30 min at 65 °C, and any contaminating RNA was removed by treating with 1 μg of RNase A (Ambion) for another 30 min at 37 °C; input material was treated equally. DNA was released by treatment with 20 μg of proteinase K (Ambion) for 4 h at 65 °C with constant shaking. DNA was purified with the QIAquick PCR purification kit (Qiagen) and eluted in 30 μL buffer EB (Qiagen). An aliquot of 1 μL each of immunoprecipitated and input DNA was analyzed in duplicate by qPCR using SYBR Green qPCR master mix (FastStart Essential DNA Green Master Mix, Roche) on a LightCycler 96 thermocycler (Roche). The relative cycle threshold (Ct) method was used to determine enrichment of immunoprecipitated DNA over input using the indicated primers; primer sequences are listed in Supplementary Data 1. Average and standard deviation values are from three biological replicates.

**5′ RACE**. The GeneRacer Kit (Thermo Scientific) was used to map 5′ ends of alternative ZFR transcripts expressed in undifferentiated THP-1 cells. In brief, 1.5 μg of total RNA was decapped with tobacco acid pyrophosphatase, and the provided RNA oligo was ligated to the 5′end of RNA with T4 RNA ligase. SuperScript III RT was used to reverse transcribe the ligated mRNA using an oligo-dT primer, and the resulting first-strand cDNA (Supplementary Fig. 3C) was amplified with a forward primer homologous to the RNA oligo and gene-specific reverse primers (Supplementary Fig. 3D, and Supplementary Data 6), and selected products were cloned and subjected to Sanger sequencing.

**ELISA**. Differentiated THP-1 cells were treated with LPS (50 ng/mL) for 6 h, and levels of secreted IFNβ was measured by VerKIne Human IFNβ ELISA kit from PBL Assay Science (ThermoFisher Scientific).

**Statistics**. Statistical tests described in the figure legends were performed using Prism software (GraphPad). $P < 0.05$ were considered significant, as detailed in the figure legends. Except for genome-wide studies, variance was similar among compared groups, and two-tailed Student's $t$-tests were performed. No data were excluded from the analyses, and sample sizes were selected based on standard procedures in the field. For differential expression studies, $P$ values were adjusted to account for multiple comparisons using DESeq2 software[58].

**Data availability**. Sequence data that support the findings of this study have been deposited in NCBI GEO with the primary accession code GSE99231.

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

## Acknowledgements

We thank Dr. John Pehrson for the macroH2A double-KO mice, Nihal Altan-Bonnet and Ying-Han Chen for providing whole-cell lysates from primary human monocytes and macrophages and for assistance with Coxsackievirus experiments, and Adrian Ferré-D'Amaré, Nico Tjandra, Richard Maraia, Nicholas Guydosh, Lisa Postow, and members of the Hogg and Ozato labs for helpful suggestions and critical reading of the manuscript. RNA-seq was performed by the NHLBI DNA Sequencing and Genomics Core, and this work utilized the computational resources of the NIH HPC Biowulf cluster. This work was supported by the Intramural Research Program, National Institutes of Health, NHLBI, NICHD.

## Author contributions

N.H. and J.R.H. initially conceived the project. N.H. performed all experiments using human cells and assays on mouse BMDM samples. R.O. and C.C conducted mouse experiments, and R.O. performed EMCV experiments. N.H., R.O., C. C., K.O., and J.R.H. designed experiments and analyzed data. N.H. and J.R.H. wrote the manuscript.

## Additional information

**Competing interests:** The authors declare no competing interests.

