## [Peer Review File(PDF 106 kb) · Nature Communications]

Reviewers' comments:

Reviewer #1 (Remarks to the Author):

Review of manuscript # NCOMMS-17-18097 entitled "ZFR coordinates crosstalk between RNA decay and transcription in innate immunity" by Hogg and coworkers. In this manuscript the authors investigate the effect of ZFR knockdown in cell lines on the expression and splicing of mRNAs in the transcriptome. They identify the mRNA encoding for the mH2A1 variant, whose expression is strongly dependent on ZFR, since knockdown of ZFR caused combined exclusion of exons 6a and 6b, which are normally included in a mutually exclusive manner. Skipping of both exons causes a frame-shift, introduces a premature stop codon and leads to nonsense-mediated mRNA decay. The expression of full-length ZFR is more prominent in differentiated macrophages, which is correlated with the expression of mH2A1, while undifferentiated THP-1 cells or BMDM express a shorter isoform, which is likely produced by an alternative promoter. The regulation of mH2A1 appears to be important for the IFN response as cell lines with knockdown of ZFR or mH2A1 produce more IFN β by increased transcription from the IFN β promoter, which itself is bound by mH2A1. The authors therefore propose model in which ZFR controls the expression of mH2A1, which can bind and repress inducible genes of innate immunity most importantly IFN β . This study provides elegant molecular mechanisms of post-transcriptional gene regulation, which are novel and of potential interest, however the manuscript does not complement these findings with in vivo evidence.

Major points

1. The specificity of the new monoclonal antibody has to be demonstrated on si-Control/siZFR treated cells (allowing the reader to judge specific from unspecific signals). An siRNA that targets mRNA sequences encoding for the N-terminus (recognizing only the full-length) and one that is located more to the 3' part of the mRNA and therefore targets both isoforms should be used. I am not convinced that (all) the shorter isoform(s) of ZFR are products of alternative transcription start sites. There could be cell-type specific protein degradation / protein cleavage events which are contributing even more strongly to the observed regulation. All Westernblots should be shown with size markers and specific bands (as determined by siRNA experiments) should be indicated.
2. Most of the experiments were conducted using 293T or THP-1 cells. Several conclusions that ZFR does not play a role in macrophage differentiation and that ZFR-dependent IFN β repression is mediated by mH2A1 in macrophages (or innate immune responses) are still questionable and require further support. The authors should employ shRNA-mediated knockdown of ZFR in the bone marrow of mice and then characterize macrophages and type I interferon responses in these bone marrow chimeric mice. Similarly, the effect of mH2A1 depletion on the IFN transcription and virus should be demonstrated in macrophages from the published mH2a1 knockout mouse.
3. In Fig. 4C quantifications and statistics on biological replicates should be provided.
4. How does overexpression of full-length and siRNA resistant cDNA of ZFR or expression of the C-terminal ZFR isoform affect the induction of IFN β and how is inappropriate ISG expression or IFN β expression prevented in undifferentiated cells.

Reviewer #2 (Remarks to the Author):

The study by Haque et al provides important insights into how a conserved zinc finger binding protein, ZFR, regulates the type I interferon response, through both RNA decay (of RNA encoding mH2A1 and promoter induction (of IFN β and possibly ISG promoters which are more inducible

in the absence of mH2A1 binding). They also propose that ZFR function is differentially regulated in undifferentiated human monocytes versus those differentiated into macrophages, through studies in THP-1 cells. This is because monocytes express a truncated variant of ZFR while macrophages show preferential expression of full length ZFR protein, due to alternative splicing. Overall the manuscript provides novel insights into the role of ZFR, and the regulation of the type I IFN response. There are some issues to address, largely to do with providing more confidence of the physiological relevance of their model:

1. The idea that macrophages and not monocytes express full length ZFR suggests that monocytes should display a heightened IFN response to pathogens compared to macrophages. Is this known to be the case? Please discuss.
2. Fig 3A - is this difference in expression of FL and truncated ZFR also apparent in primary human monocytes versus macrophages? It would be important to test this, since THP-1 cells, although often used as a model of human monocytes, do display some unusual attributes in terms of regulation of innate immune genes.
3. Fig 5C - the authors should also show an ELISA for IFNbeta, or a type I IFN bioassay in order to demonstrate that regulation of promoter-proximal events by ZFR do in fact affect a real IFN response.
4. In order to really prove that truncated ZFR and full length ZFR display different effects on type I IFN responses, the authors should rescue THP-1 cells deficient in ZFR with full length versus truncated ZFR and compare the IFN β mRNA responses after LPS or virus treatment (Fig 5C).
5. Fig 6I - Is there any stimulus-dependent (e.g. LPS, pIC) regulation of mH2A1 occupancy of the IFNbeta locus? Or is the model that in macrophages mH2A1 is constitutively present, and not further recruited after stimulation? Also can the authors make a meaningful comparison of the occupancy of the IFNbeta locus by mH2A1 in undifferentiated versus differentiated THP-1s, in order to correlate with the lack of full length (ie active) ZFR in monocytes?

We thank the reviewers for their comments, which we believe have helped us to substantially strengthen the manuscript. Please find our detailed responses below.

Reviewers' comments:

Reviewer #1 (Remarks to the Author):

Review of manuscript # NCOMMS-17-18097 entitled "ZFR coordinates crosstalk between RNA decay and transcription in innate immunity" by Hogg and coworkers. In this manuscript the authors investigate the effect of ZFR knockdown in cell lines on the expression and splicing of mRNAs in the transcriptome. They identify the mRNA encoding for the mH2A1 variant, whose expression is strongly dependent on ZFR, since knockdown of ZFR caused combined exclusion of exons 6a and 6b, which are normally included in a mutually exclusive manner. Skipping of both exons causes a frame-shift, introduces a premature stop codon and leads to nonsense-mediated mRNA decay. The expression of full-length ZFR is more prominent in differentiated macrophages, which is correlated with the expression of mH2A1, while undifferentiated THP-1 cells or BMDM express a shorter isoform, which is likely produced by an alternative promoter. The regulation of mH2A1 appears to be important for the IFN response as cell lines with knockdown of ZFR or mH2A1 produce more IFN β by increased transcription from the IFN β promoter, which itself is bound by mH2A1. The authors therefore propose model in which ZFR controls the expression of mH2A1, which can bind and repress inducible genes of innate immunity most importantly IFN β .

This study provides elegant molecular mechanisms of post-transcriptional gene regulation, which are novel and of potential interest, however the manuscript does not complement these findings with in vivo evidence.

Major points

1. The specificity of the new monoclonal antibody has to be demonstrated on si-Control/siZFR treated cells (allowing the reader to judge specific from unspecific signals). An siRNA that targets mRNA sequences encoding for the N-terminus (recognizing only the full-length) and one that is located more to the 3' part of the mRNA and therefore targets both isoforms should be used. I am not convinced that (all) the shorter isoform(s) of ZFR are products of alternative transcription start sites. There could be cell-type specific protein degradation / protein cleavage events which are contributing even more strongly to the observed regulation. All Western blots should be shown with size markers and specific bands (as determined by siRNA experiments) should be indicated.

In order to demonstrate the specificity of the monoclonal antibody, cell lysates from HEK293 cells stably expressing shRNA against ZFR or scrambled shRNA were used for western

blotting. Specificity of the antibody can be assessed from the western blot analysis (Fig.5E), which shows clear signal from full-length protein in control cells but not in shZFR cells. We agree that the presence of multiple bands in ZFR immunoblots of monocytes shown in the initial submission complicated the interpretation of our data. In the course of performing experiments for the revisions, we realized that the short isoform of ZFR is highly susceptible to disulfide crosslinking if not treated with fresh reducing agents immediately prior to gel electrophoresis. With proper reduction conditions, we found that the predominant doublet corresponding to the small isoform migrated more quickly than the major truncated isoforms detected in our earlier gels (Fig. 3A). In addition, many of the fainter bands present in prior western blots of undifferentiated THP-1 cells appear to have collapsed into this doublet. We provide further evidence for differential TSS usage in monocytes and macrophages by performing 5'-RACE using total RNA from both cell states (Supplementary Fig. 3C). A cDNA species of approximately 1.0 Kb was found to be abundant in undifferentiated cells and decreased substantially upon differentiation. As shown in the initial submission, cloning and sequencing of the smaller cDNA suggested that this isoform is generated by usage of an alternative promoter. We have now expressed the small isoform from these cDNAs, producing proteins with similar mobility to those expressed in monocytic THP-1 cells (compare Fig. 3A with Supplementary Fig. 4C). Moreover, we mined the FANTOM/RIKEN CAGE-seq data, which reports a cluster of alternative TSSs in the vicinity of the beginning of the 5'RACE product (Supplementary Fig. 4A). Strikingly, FANTOM analysis of hundreds of CAGE-Seq experiments from numerous human primary cell types and cell lines finds that this cluster of alternative promoters is preferentially used in monocytic cells (available on the ZENBU browser and presented in Supplementary Fig. 4A). Importantly, while the occurrence of alternative promoters is conserved in mice, per FANTOM CAGE-Seq (Supplementary Fig. 4B), their positioning is different. Under fully reducing conditions, the bands detected by ZFR immunoblotting in mouse BMDMs match the predicted protein sizes based on the location of the alternative mouse TSSs. Together, we present several independent lines of evidence indicating that long and short isoforms are generated using different promoters, but we also acknowledge in the revised discussion that other mechanisms such as proteolytic cleavage or protein degradation may also aid in producing differential ZFR isoforms.

2. Most of the experiments were conducted using 293T or THP-1 cells. Several conclusions that ZFR does not play a role in macrophage differentiation and that ZFR-dependent IFN β repression is mediated by mH2A1 in macrophages (or innate immune responses) are still questionable and require further support. The authors should employ shRNA-mediated knockdown of ZFR in the bone marrow of mice and then characterize macrophages and type I interferon responses in these bone marrow chimeric mice. Similarly, the effect of mH2A1 depletion on the IFN transcription and virus should be demonstrated in macrophages from the published mH2a1 knockout mouse.

To show the physiological effect of macroH2A on macrophage differentiation and the IFN response, we obtained previously reported macroH2A double-knockout mice. BMDMs isolated from the knockout or WT mouse were differentiated to macrophages and stimulated with poly I:C and LPS to induce IFN β . Several-fold higher induction of IFN β , as well as downstream ISGs were observed in the macroH2A knockout BMDMs compared to wild-type cells (Fig. 6K, L, Supplementary Fig. 7H-J), consistent with our knockdown data in HEK293 cells and in vitro RNAi in WT BMDMs (Fig. 6E, G, Supplementary Fig. 7F).

We did not intend to strongly argue against a role for ZFR in macrophage differentiation based on our data, but instead wished to provide evidence that the splicing changes observed in the ZFR knockdown cells are unlikely to be due to a differentiation defect per se. We have revised the text to more clearly communicate this point. While we consider a further exploration of the role of ZFR in macrophage differentiation an interesting subject for future work, we believe that such studies are beyond the scope of this manuscript.

3. In Fig. 4C quantifications and statistics on biological replicates should be provided.

Analysis from three biological replicates was included for all of the splicing targets shown in Fig. 4C, with the exception of Fyn, for the technical reason that it was not possible to sufficiently separate the isoforms (which differ by only 9 nt) to obtain reliable quantification. Fyn shows a strong response to ZFR depletion in triplicate RNAseq samples in both HEK293 and THP-1 cells, so we remain confident in its identification as a ZFR-regulated splicing event (Fig. 4B, C).

4. How does overexpression of full-length and siRNA resistant cDNA of ZFR or expression of the C-terminal ZFR isoform affect the induction of IFN β and how is inappropriate ISG expression or IFN β expression prevented in undifferentiated cells.

As we show that ZFR's primary effect on IFN β production is a result of aberrant macroH2A1 splicing, we focused our inquiries on the ability of full-length and truncated ZFR to rescue macroH2A1 mRNA expression following endogenous ZFR depletion. We transiently transfected ZFR-depleted cells with shRNA-resistant short or full-length ZFR cDNAs and assayed for rescue of macroH2A1 mRNA. Efficient rescue of macroH2A1 mRNA was detected in cells transfected with full-length cDNA, but no change in macroH2A1 mRNA was observed with expression of the smaller isoform in ZFR-depleted cells (Supplementary Fig. 4C, D). Taken together, we conclude that the regulation of macroH2A1 and subsequent relaxation of IFN β repression is mediated by full-length ZFR, and our experiments thus far do not suggest either a positive or negative role for the truncated protein in this process. However, we do not rule out that the truncated isoform may have as-yet-undetected activities that influence innate immune responses. Due to inefficient transfection of THP-1

cells, detailed studies of any such additional roles will require extensive further development of knockout and overexpression lines in THP-1 cells.

Reviewer #2 (Remarks to the Author):

The study by Haque et al provides important insights into how a conserved zinc finger binding protein, ZFR, regulates the type I interferon response, through both RNA decay (of RNA encoding mH2A1 and promoter induction (of IFN β and possibly ISG promoters which are more inducible in the absence of mH2A1 binding). They also propose that ZFR function is differentially regulated in undifferentiated human monocytes versus those differentiated into macrophages, through studies in THP-1 cells. This is because monocytes express a truncated variant of ZFR while macrophages show preferential expression of full length ZFR protein, due to alternative splicing. Overall the manuscript provides novel insights into the role of ZFR, and the regulation of the type I IFN response. There are some issues to address, largely to do with providing more confidence of the physiological relevance of their model:

1. The idea that macrophages and not monocytes express full length ZFR suggests that monocytes should display a heightened IFN response to pathogens compared to macrophages. Is this known to be the case? Please discuss.

Macrophage differentiation is accompanied by induction of many components of the innate immune system, including proteins responsible for IFN β induction such as IRF transcription factors and TLRs. This suggests that monocytes, despite expressing the short isoform of ZFR and having lower levels of macroH2A1, may not produce a heightened immune response compared to macrophages. Instead, we speculate that one reason macrophages induce ZFR and macroH2A expression is to dampen the heightened sensitivity to PAMPs caused by increased expression of TLRs and other innate immune factors. This hypothesis is difficult to test due to the many differences between monocytes and macrophages, but we hope to make progress toward this in the future.

2. Fig 3A - is this difference in expression of FL and truncated ZFR also apparent in primary human monocytes versus macrophages? It would be important to test this, since THP-1 cells, although often used as a model of human monocytes, do display some unusual attributes in terms of regulation of innate immune genes.

We performed western blotting on lysates from primary human monocytes and differentiated macrophages, finding that they also undergo differential expression of truncated and full-length ZFR isoforms (Supplementary Fig. 3B). We note that in addition to isoforms that co-migrate with the major truncated isoform from monocytic THP-1 cells,

primary monocytes also show a second smaller isoform of unknown origin. Most importantly, all three settings in which we have tested ZFR expression in macrophage differentiation (mouse BMDMs, THP-1 cells, and human primary cells) show low expression of full-length ZFR in monocytes, with accompanying appearance of smaller isoforms.

3. Fig 5C - the authors should also show an ELISA for IFN β , or a type I IFN bioassay in order to demonstrate that regulation of promoter-proximal events by ZFR do in fact affect a real IFN response.

Secreted levels of IFN β were measured by ELISA after treating ZFR-depleted or control THP-1 cells with LPS (Fig 5D). In addition, Fig. 6H, Supplementary Fig. 6B, and Supplementary Fig. 7H-J show induction of downstream ISGs, consistent with the enhanced levels of IFN β in ELISA experiments.

4. In order to really prove that truncated ZFR and full length ZFR display different effects on type I IFN responses, the authors should rescue THP-1 cells deficient in ZFR with full length versus truncated ZFR and compare the IFN β mRNA responses after LPS or virus treatment (Fig 5C).

As described in the response to reviewer 1 (point 4), ZFR-depleted HEK-293 cells were transiently transfected with short or full-length ZFR cDNAs and assessed for rescue of mH2A1 mRNA, as our data indicate that this is central to IFN β repression (Supplementary Fig. 4 C and D). More than 10-fold higher levels of mH2A1 mRNA were detected in ZFR-depleted cells transfected with full-length ZFR cDNA compared to GFP control. There was no significant change in mH2A1 mRNA level upon overexpression of the short isoform of ZFR.

5. Fig 6I - Is there any stimulus-dependent (e.g. LPS, pIC) regulation of mH2A1 occupancy of the IFN β locus? Or is the model that in macrophages mH2A1 is constitutively present, and not further recruited after stimulation? Also can the authors make a meaningful comparison of the occupancy of the IFN β locus by mH2A1 in undifferentiated versus differentiated THP-1s, in order to correlate with the lack of full length (ie active) ZFR in monocytes?

No significant change in macroH2A1 occupancy was observed at the IFN β promoter with or without stimulation with poly I:C (HEK-293 cells; Supplementary Fig. 7G), suggesting that macroH2A1 is constitutively present at the IFN β locus. This is consistent with previous findings (for example see Gamble et al., G&D, 2010) that macroH2A's effect on inducible promoters is not necessarily accompanied by changes in occupancy. We have considered comparing mH2A1 occupancy of the IFN β promoter in monocytic and macrophage-like

THP-1 cells, but the overall properties and gene expression profiles between differentiated and undifferentiated cells are substantially distinct, making any quantitative comparisons from ChIP studies unreliable.

REVIEWERS' COMMENTS:

Reviewer #1 (Remarks to the Author):

The authors have addressed my concerns in depth.

Reviewer #2 (Remarks to the Author):

The authors have addressed all the reviewers concerns adequately.